# TRAINING-FREE STYLIZED ABSTRACTION

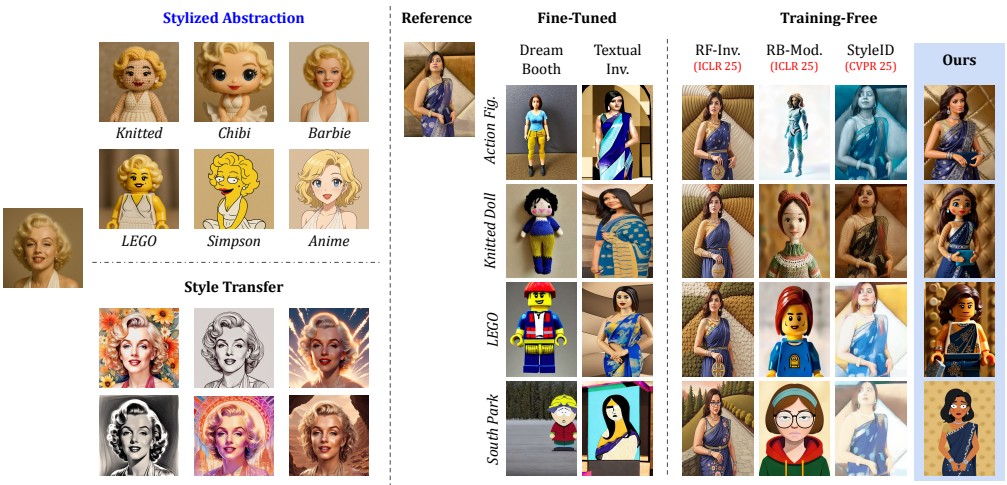

Figure 1: **a)** Style Abstraction vs. Traditional style transfer. **(Top)** Stylized abstraction techniques capture core identifying attributes while allowing stylistic distortion to preserve the intended visual style. **(Bottom)** Traditional style transfer preserves geometry and appearance but applies texture-based styles, often failing to generalize beyond appearance-level edits. **b)** Comparison across existing style transfer/personalized generation using a **single image** of a **non-celebrity subject**. Most methods struggle to retain semantic identity for everyday individuals, while our **training-free** method preserves key identity cues across diverse styles.

## ABSTRACT

Stylized abstraction synthesizes visually exaggerated yet semantically faithful representations of subjects, balancing recognizability with perceptual distortion. Unlike image-to-image translation, which prioritizes structural fidelity, stylized abstraction demands selective retention of identity cues while embracing stylistic divergence, especially challenging for out-of-distribution individuals. We propose a training-free framework that generates stylized abstractions from a single image using inference-time scaling in vision-language models (VLLMs) to extract identity-relevant features, and a novel cross-domain rectified flow inversion strategy that reconstructs structure based on style-dependent priors. Our method adapts structural restoration dynamically through style-aware temporal scheduling, enabling high-fidelity reconstructions that honor both subject and style. It supports multi-round abstraction-aware generation without fine-tuning. To evaluate this task, we introduce *StyleBench*, a GPT-based human-aligned metric suited for abstract styles where pixel-level similarity fails. Experiments across diverse abstraction (e.g., LEGO, knitted dolls, South Park) show strong generalization to unseen identities and styles in a fully open-source setup.

## 1    INTRODUCTION

**Image-to-image style translation** Deng et al. (2022); Sohn et al. (2023); Wang et al. (2023); Jiang & Chen (2024); Jing et al. (2019); Xing et al. (2024); Chen et al. (2021) is a well-studied area that traces

its origins to GAN-based approaches, such as neural style transfer Gatys et al. (2015) and CycleGAN Zhu et al. (2017), and has since evolved to include both diffusion-based training and training-free methods Rout et al. (2025a;b); Le & Carlsson (2022); Mo et al. (2024); Zhang et al. (2023); Zhao et al. (2023); Deng et al. (2022); Liu et al. (2023a); Chen et al. (2024). These techniques typically focus on overlaying a specific style onto an input image while preserving the subject's identity. Common examples include transforming portraits into sketches, cartoons, or artwork in the style of artists like Van Gogh. Importantly, the resulting stylized images often retain structural consistency with the original content.

**Stylized abstraction**, on the other hand, involves exaggerating or simplifying the features of a subject to create a stylized representation. Rather than aiming for photorealism, it emphasizes recognizable traits that evoke the subject's **concept or identity** (Illustrated in Figure 1 (a)). Stylized representation aims to capture the essence of a subject through *visual abstraction*, focusing less on exact likeness and more on the retention of key, recognizable features Berger et al. (2013). For instance, a knitted doll or a LEGO figure of Einstein may omit intricate facial geometry or biometric precision, yet still be immediately identifiable due to consistent visual traits such as his distinctive hair, mustache, or attire. These features serve as semantic anchors, allowing viewers to recognize the subject even in highly abstracted or playful forms. This form of representation is widespread in media, animation, and merchandising, where retaining a character's identity in a simplified, reproducible form is essential. Terms like *personified toy representation* or *iconic stylization* are often used to describe such instances. Unlike traditional image-to-image translation, which typically enforces structural consistency, stylized abstraction embraces simplification, distortion, or even exaggeration to evoke familiarity and conceptual identity.

Stylized abstraction, in contrast to traditional image-to-image translation, remains a relatively under-explored topic. The challenge lies in its nuanced nature. While image-to-image translation typically involves aligning an input image with the distribution of a target style, often while preserving geometric structure. This is a relatively simpler task. For example, sketches can be viewed as stylized edge maps, or many artistic styles merely impose brushstroke patterns and color palettes onto the input image. Stylized abstraction, however, demands more than stylistic transfer; it requires a careful balance between simplification and recognizability. It involves distilling the subject to its most *iconic traits*, sometimes exaggerating them while discarding fine-grained details. This abstraction introduces greater semantic and structural deviation from the input, making the problem far more complex than merely applying texture or color-based transformations.

Now, while a number of image stylization methods exist ranging from training-free techniques to fine-tuning-based pipelines Ruiz et al. (2023); Gal et al. (2022) or encoder-based methods Li et al. (2023); Ye et al. (2023) that adapt reference features for concept preservation in general T2I models these approaches often fall short in the domain of stylized abstraction. Notably, many existing works demonstrate results primarily on celebrity faces, which are already well-represented in pre-trained models. As a result, these models often succeed in retaining recognizable features simply because they have been exposed to those identities during training. However, when tested on images of everyday individuals, these same methods either fail to preserve identity or compromise the intended stylization (Visualized in Figure 1 (b)). This highlights the need for methods that can generalize stylized abstraction to diverse identities without relying on prior memorization.

To address the limitations of existing stylization methods, we present a training-free framework for stylized abstraction that generalizes beyond celebrity likenesses to everyday identities and supports a wide range of abstract styles—such as LEGO, South Park, Simpson, Matrushka, Barbie, Knitted Doll, and Action Figure, **without relying on any predetermined or fixed set of styles. These examples are illustrative rather than exhaustive, and our method flexibly adapts to new, unseen styles at inference time without requiring retraining.** Our approach requires neither subject-specific fine-tuning nor dataset-level adaptation. Instead, we introduce a novel inference-time scaling strategy for vision-language models (VLLMs) that distills core semantic traits critical for identity preservation and aligns them with user-driven stylistic prompts. Central to our pipeline is a multi-turn generation loop, where missing or distorted identity cues, identified via VLLM feedback, are progressively reintegrated to enhance fidelity across iterations. To recover subject structure under extreme abstraction, we extend RF-Inversion Rout et al. (2025a) with a cross-domain latent inversion scheme, treating stylized images as source latents and photorealistic representations as structural targets. Leveraging rectified flow-guided updates and style-aware temporal scheduling, our method preserves stylistic fidelity while selectively restoring identity-consistent structure in a controllable and

interpretable fashion. To evaluate abstraction beyond pixel-level similarity, we introduce StyleBench, a GPT-assisted, human-aligned protocol for benchmarking stylized abstraction. We further report quantitative performance using KID and CLIPScore, supported by a user preference study. Our framework sets a new state-of-the-art in abstraction quality: fully training-free, identity-consistent, and broadly generalizable across styles and subjects.

## 2 RELATED WORK

**Identity-Preserving Style Transfer.** Identity-preserving or subject-driven style transfer Zhang et al. (2024); Raj et al. (2023); Chen et al. (2023b); Miao et al. (2024); Dong et al. (2022); Han et al. (2023a); Voynov et al. (2023); Alaluf et al. (2023); Kumari et al. (2023); Liu et al. (2023b); Han et al. (2023b); Ryu (2023); Avrahami et al. (2023); Chen et al. (2023a); Cai et al. (2024); He et al. (2025) is a closely related line of work to stylized abstraction, where the goal is to synthesize stylized images while retaining subject identity. Approaches in this domain broadly fall into three categories. The first category includes fine-tuning-based methods, such as Textual Inversion Gal et al. (2022) and DreamBooth Ruiz et al. (2023), which adapt the generative model to the target subject using multiple reference images. While effective for object-centric domains, these methods often struggle to faithfully preserve identity in human subjects and require several input images for fine-tuning. The second category comprises encoder-based methods that learn feature adaptation modules to modulate the base model without explicit fine-tuning. Notable examples include IP-Adapter Ye et al. (2023) and BLIP-Diffusion Li et al. (2023), which leverage pretrained encoders to align content and style representations. The third category focuses on single-image personalization, including DreamTuner Hua et al. (2023) and CSGO Xing et al. (2024), or entirely training-free techniques such as RF-Inversion Rout et al. (2025a), RB-Modulation Rout et al. (2025b), StyleID Le & Carlsson (2022), InstantID Wang et al. (2024b), InstantID-Plus Wang et al. (2024a), and DiffArtist Jiang & Chen (2024). These methods often rely on CLIP-guided optimization or feature injection to steer generation toward the desired style. However, such methods typically prioritize structural fidelity or stylization strength in isolation. In high stylization scenarios, they may struggle to abstract and reinterpret content meaningfully, as their architectures tend to enforce either identity preservation or stylistic consistency without deeper semantic understanding. Bridging this gap requires novel approaches that reason over semantic correspondences between style and identity, rather than relying solely on pixel or feature-space alignment.

**Multi-Modal LLMs in Personalized Image Generation.** Recent advances in multi-modal large language models (MLLMs) have demonstrated their potential in various image generation tasks Gal et al. (2022); Liu et al. (2025); Liao et al. (2024); Wu et al. (2024b); Sun et al. (2024); Wu et al. (2024a), although not always directly targeting personalized image generation. These models exhibit strong generalization capabilities when applied to complex and previously unseen scenarios Hu et al. (2024); Qu et al. (2023); Wang et al. (2024c). Leveraging both multi-modal understanding and generative modeling, commercial models such as GPT-4o OpenAI (2024), Gemini DeepMind (2023), and Grok xAI (2025) have recently shown the ability to produce stylized and personalized images from user inputs. However, we highlight several limitations in this emerging line of work. Most of these systems are proprietary and closed-source, trained on large-scale datasets that are not publicly available. Furthermore, it remains unclear whether the outputs involve additional fine-tuning or personalization modules beyond the core model. These factors hinder reproducibility and limit academic scrutiny.

## 3 METHOD

### 3.1 SUBJECT IDENTITY DISTILLATION VIA INFERENCE-TIME VLLM SCALING

**Dense Attribute Extraction.** Given an input image $\mathbf{I} \in \mathbb{R}^{H \times W \times 3}$, we initiate a multi-round interaction with a vision-language language model (VLLM) Zhu et al. (2025) to obtain exhaustive descriptions of identity-related features. Let $\mathcal{V}$ denote the VLLM interface. The process is divided into four semantically disjoint rounds: facial attributes ($\mathcal{A}_{\text{face}}$), clothing and accessories ($\mathcal{A}_{\text{attire}}$), posture and pose ($\mathcal{A}_{\text{pose}}$), and background environment ($\mathcal{A}_{\text{scene}}$). Each round is conditioned on $\mathbf{I}$ and prompts $\mathcal{P}_k$ specifically tailored to query salient features of category $k$:

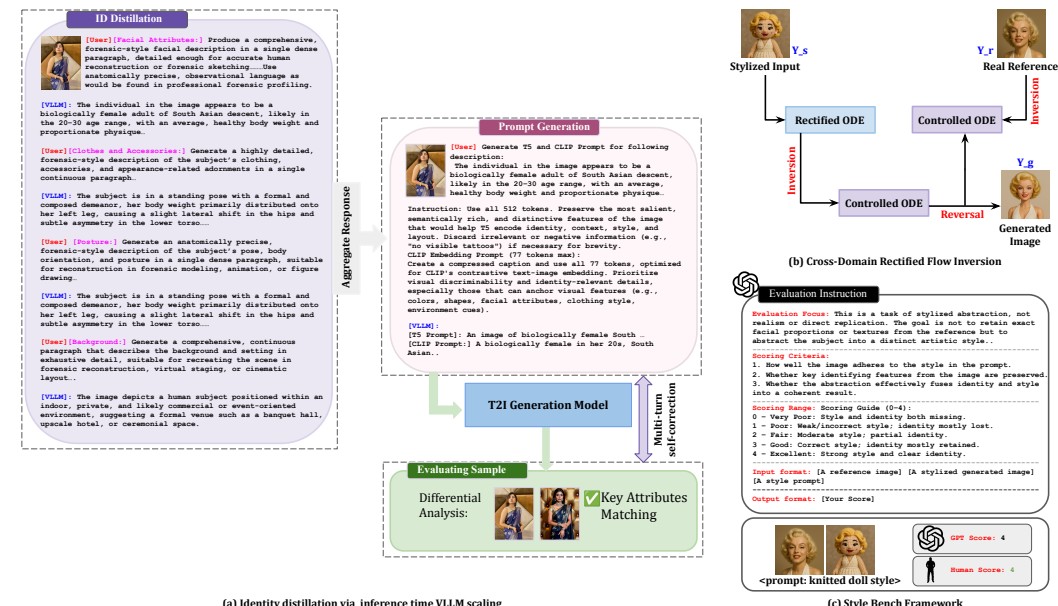

Figure 2: (a) **Workflow of identity distillation via inference-time VLLM scaling.** The process includes dense attribute extraction, multi-scale prompt compression, iterative identity refinement, and style-aware prompt transformation. (b) **Cross-domain Latent Reversal** pipeline for stylized image generation. (c) End-to-end workflow of the **StyleBench** evaluation framework.

$$\mathcal{A}_k = \mathcal{V}(\mathbf{I}, \mathcal{P}_k), \quad \text{for } k \in \{\text{face}, \text{attire}, \text{pose}, \text{scene}\}. \tag{1}$$

The outputs $\{\mathcal{A}_k\}$ are structured natural language descriptions optimized for visual grounding.

**Multi-Scale Prompt Compression.** The extracted descriptions are aggregated and passed to a secondary VLLM instance $\mathcal{V}'$, which synthesizes two task-specific prompts- a) $\mathcal{T}_{512}$: A 512-token prompt optimized for T5-based Raffel et al. (2020) generators, preserving identity, context, style, and layout. b) $\mathcal{T}_{77}$: A 77-token CLIP-style Radford et al. (2021) prompt, distilled to maximize contrastive relevance in embedding space.

Formally, let $\mathcal{A}_{\text{full}} = \bigcup_k \mathcal{A}_k$, then:

$$\mathcal{T}_{512}, \mathcal{T}_{77} = \mathcal{V}'(\mathcal{A}_{\text{full}}). \tag{2}$$

**Iterative Identity Refinement.** The condensed prompts $\mathcal{T}_{512}, \mathcal{T}_{77}$ are used as conditioning inputs to an image generation pipeline, Flux Labs (2024), producing a candidate image $\hat{\mathbf{I}}$. A third VLLM instance $\mathcal{V}''$ performs a differential analysis between the original image $\mathbf{I}$ and generated image $\hat{\mathbf{I}}$, identifying missing or misaligned attributes:

$$\Delta\mathcal{A} = \mathcal{V}''(\mathbf{I}, \hat{\mathbf{I}}). \tag{3}$$

These attributes $\Delta\mathcal{A}$ are incrementally reintegrated into the textual representation by updating $\mathcal{A}_{\text{full}} \leftarrow \mathcal{A}_{\text{full}} \cup \Delta\mathcal{A}$, prompting a regeneration of $\mathcal{T}_{512}, \mathcal{T}_{77}$. This loop continues until either a perceptual alignment threshold is reached (e.g., $\text{CLIP}(\mathbf{I}, \hat{\mathbf{I}}) \geq \tau$) or a maximum number of rounds $T$ is completed.

**Inference-Time Identity Convergence.** This inference-time distillation strategy enables progressive identity preservation without requiring gradient updates. The architecture remains fixed; only VLLM feedback adaptively steers prompt construction. The convergence criterion is defined as:

$$\text{stop} \iff \text{CLIP}(\mathbf{I}, \hat{\mathbf{I}}^{(t)}) \geq \tau \quad \text{or} \quad t \geq T, \tag{4}$$

where $\hat{\mathbf{I}}^{(t)}$ is the generated image at iteration $t$.

This multi-turn VLLM-in-the-loop mechanism emulates a self-correcting distillation process, bridging perceptual gaps between the source image and its identity representation without paired supervision.

**Style-Aware Prompt Transformation.** The updated prompt pair $(\mathcal{T}512, \mathcal{T}77)$ undergoes a style-conditioning step. A final VLLM module $\mathcal{V}^*$ is invoked with a style descriptor $\mathcal{S}$ (e.g., "knitted doll", "LEGO", or "anime") to adapt the identity-rich prompt into a stylized version while preserving semantic fidelity:

$$\mathcal{T}^{\text{styled}}512, \mathcal{T}^{\text{styled}}77 = \mathcal{V}^*(\mathcal{T}512, \mathcal{T}77, \mathcal{S}). \tag{5}$$

These stylized prompts guide the Flux generation pipeline to produce an initial abstraction $y_s$. An overview of the framework is shown in Figure 2 (a).

## 3.2 CROSS-DOMAIN LATENT REVERSAL WITH RECTIFIED FLOWS

At this stage, we obtain a highly stylized representation of the subject with faithfully preserved stylistic elements, $y_s$, but the generation now requires structural guidance from the original image to recover key identity-aligned geometry. However, unlike prior inversion-based pipelines Rout et al. (2025a) that operate on realistic or noisy inputs, our setting begins from an already abstracted, stylized image $y_s$. This shift introduces a novel challenge: how to reconstruct a semantically grounded latent representation from a highly altered input while flexibly recovering structural details based on style demands.

To address this, we propose a two-stage framework for cross-domain latent reversal, which extends rectified flow methods Rout et al. (2025a) into the abstract stylization space. The key is to treat the stylized abstraction not as a degraded variant of a photo, but as a valid starting point in an altered visual domain: one that must be softly regularized back into a structured latent aligned with the true subject identity.

In the first stage, we invert the stylized image $y_s$ using a forward rectified ODE:

$$dY_t = [u_t(Y_t) + \gamma\left(u_t(Y_t \mid y_1) - u_t(Y_t)\right)] dt, \quad Y_0 = y_s, \tag{6}$$

where $u_t(Y_t)$ denotes the unconditional drift field from the pretrained Flux model, and $u_t(Y_t \mid y_1)$ is an analytically derived controller via linear quadratic regulation (LQR). The scalar $\gamma \in [0, 1]$ governs the balance between staying close to the stylized abstraction and conforming to the learned noise prior $y_1 \sim \mathcal{N}(0, I)$. This inversion step introduces the application of rectified flows to high-level stylistic abstractions, where the latent does not correspond to a photo-realistic image but to a semantically enriched domain-specific representation.

The second stage performs structure-aware reconstruction using a controlled reverse ODE, initialized from $y_1$ and guided toward a real reference image $y_r$:

$$dX_t = [v_t(X_t) + \eta_t\left(v_t(X_t \mid y_r) - v_t(X_t)\right)] dt, \quad X_0 = y_1, \tag{7}$$

where $v_t$ is the reverse-time vector field. The time-varying structural controller $\eta_t$ defined as:

$$\eta_t = \begin{cases} \eta, & t \in [\tau_{\text{start}}, \tau_{\text{stop}}] \\ 0, & \text{otherwise} \end{cases} \tag{8}$$

Unlike fixed-strength guidance, this scheduling allows us to inject structural constraints only during a **style-dependent** temporal window. Such a design is critical for abstract styles where early over-regularization can collapse style integrity. Parameters $\eta$, $\tau_{\text{start}}$, and $\tau_{\text{stop}}$ are adaptively chosen using a VLLM-based controller that parses the style descriptor (e.g. "knitted doll", "South Park") to determine the structural necessity. Overview of the process is shown in Figure 2 (b).

### 3.3 STYLEBENCH: HUMAN-ALIGNED EVALUATION FOR STYLIZED ABSTRACTION

Evaluating stylized image generation requires going beyond traditional notions of visual similarity. In many artistic or character-driven styles, such as knitted dolls, LEGO figures, or South Park characters, the geometry, texture, and proportions of the original subject are intentionally distorted. However, what remains essential is the preservation of key identity cues: hairstyle, posture, clothing, or accessories that allow recognition despite abstraction. Existing benchmarks like DreamBench++Peng et al. (2024) have made progress in human-aligned evaluation for personalized image generation, particularly by assessing prompt consistency and subject fidelity using multimodal GPT-4o model. However, these benchmarks primarily operate under assumptions of semantic and structural coherence typical of photorealistic or lightly stylized domains.

In contrast, *StyleBench* is tailored specifically for *stylized abstraction*, where the visual transformation is often extreme and the identity must be reinterpreted through a unique stylistic lens. Our benchmark introduces a structured evaluation protocol using GPT models, which are prompted with three inputs: a reference image, a stylized generation, and a style prompt as shown in Figure 2 (c). The task definition guides the model to assess not realism or one-to-one replication, but how well the abstraction balances fidelity to the subject's recognizable identity with faithful adherence to the style's visual language.

To ensure consistent and human-aligned evaluation, we design the GPT prompt with explicit scoring criteria across three integrated axes: (i) adherence to style, (ii) identity preservation, and (iii) fusion quality. The evaluation process incorporates internal task summarization and optional chain-of-thought reasoning to encourage self-alignment Peng et al. (2024); Sun et al. (2023) before issuing a score between 0 (very poor) and 4 (excellent). Unlike generic perceptual metrics (e.g., CLIP Radford et al. (2021), DINO Zhang et al. (2022)), which often fail under stylization shifts, our protocol enables nuanced judgments aligned with how humans interpret abstracted identity. This makes *StyleBench* particularly suitable for benchmarking models that target stylized avatars, artistic reinterpretations, and toy-based renderings, domains where abstraction is not a flaw but a defining feature.

## 4 EXPERIMENT

**Baselines** There is no direct baseline for stylized abstraction, as it is a relatively new concept in the vision community that goes beyond identity preservation to include semantic and geometric reinterpretation. We compare against the closest related methods across personalization and style transfer. These include fine-tuning-based approaches such as Textual Inversion Gal et al. (2022) and DreamBooth Ruiz et al. (2023), encoder-based CSGO Xing et al. (2024), and training-free, zero-shot methods including StyleID Le & Carlsson (2022), RF-Inversion Rout et al. (2025a), RB-Modulation Rout et al. (2025b), DiffArtist Jiang & Chen (2024), InstantID Wang et al. (2024b), and InstantID-Plus Wang et al. (2024a).

**Dataset and Evaluation metric.** Our dataset consists of three categories: (i) single subject images of everyday individuals, comprising 10 images across 10 unique subjects; (ii) multi-subject images of everyday individuals, totaling 14 images; and (iii) single-subject celebrity images collected from Google Image Search under free use licenses, amounting to 30 images. For evaluation, we employ Kernel Inception Distance (KID) Bińkowski et al. (2018), CLIP score, our proposed StyleBench benchmark, and human evaluation. The human evaluation is conducted on 25 generated images, rated by 15 independent annotators.

**Implementation Details** We implement all models using PyTorch and run experiments on NVIDIA A6000. We employ InternVL Zhu et al. (2025) as the VLLM and FLUX Labs (2024) as the image generation backbone. More details on baseline reproduction are provided in the supplementary.

## 5 RESULTS AND ANALYSIS

**Qualitative Results.** Figure 3 presents a diverse set of stylized abstractions across 10 styles, including regional representations and balanced gender coverage. Baseline methods often fail because they are not specifically designed for high-level abstraction tasks, particularly in training-free settings with only a single reference image. These models typically lack mechanisms to semantically disentangle identity from style, leading to poor content preservation or shallow stylistic transfer. On the other

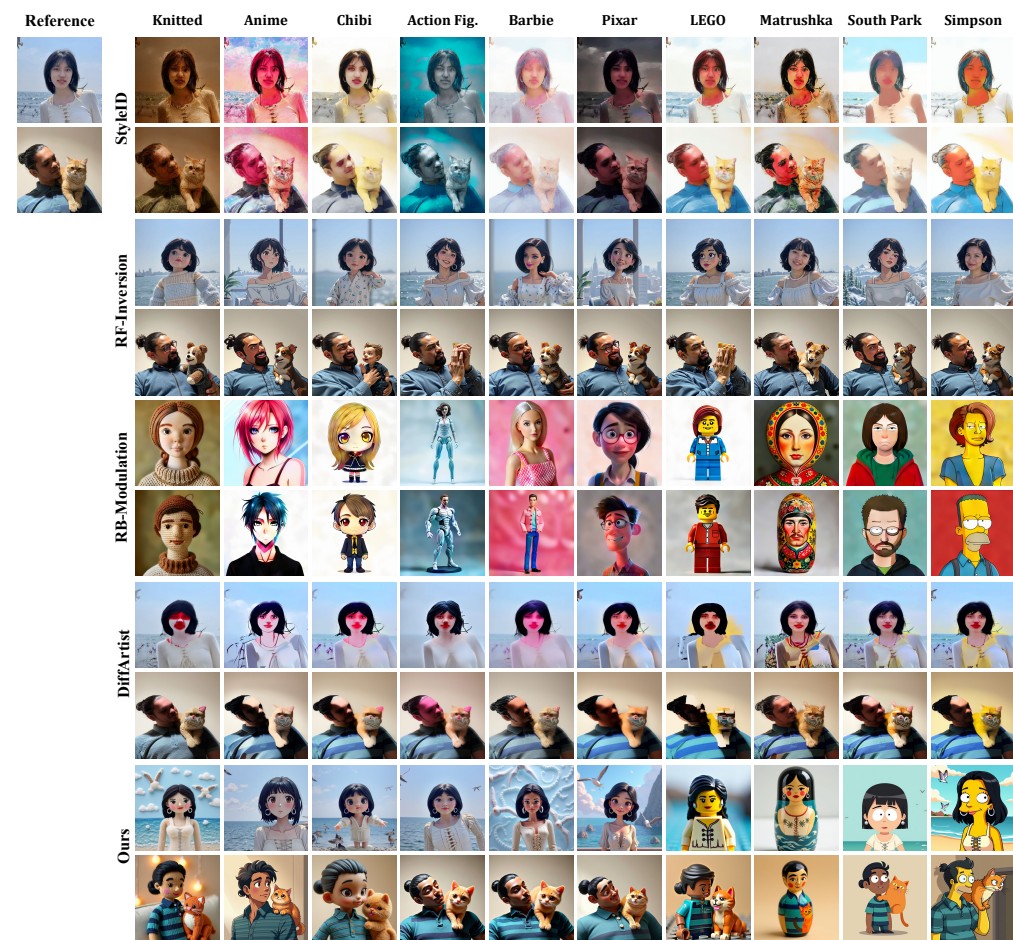

Figure 3: **Qualitative comparison with existing image stylization models.** Most prior methods struggle to preserve either the reference content or the intended style. For example, models like StyleID Le & Carlsson (2022) rely on a reference style image and often perform low-level pixel blending, which fails to generalize to high-level abstractions. In contrast, our method preserves both identity and style with high semantic fidelity.

hand, our method consistently retains subject essence while embracing the stylistic exaggeration unique to each domain. Some additional results with more concepts are shown in Figure 5.

**Note.** While we primarily present portrait stylization results, this choice reflects the fact that human identity preservation under abstraction is among the most challenging scenarios. Methods such as RF-Inversion and RB-Modulation generalize well to everyday objects (e.g., landmarks, pets, fruits), where semantic identity is far less complex. For instance, turning a tomato into a knitted abstraction is comparatively trivial, whereas ensuring that a knitted version of a person still retains recognizable identity is substantially harder. Hence, we focused our evaluation on human portraits to highlight the difficulty of the problem rather than the ease of style transfer in simpler domains.

**Quantitative Results.** Table 1 compares our method against existing baselines across KID, CLIP score, StyleBench, and human evaluation. Our method achieves competitive performance across all metrics, particularly excelling in human-aligned scores, highlighting its ability to produce abstractions that are both recognizable and stylistically faithful.

**Impact of Identity-Distilled Prompts.** We investigate the effect of using dense, identity-distilled prompts obtained via inference-time querying of a VLLM to extract subject-specific attributes. This experiment evaluates how effectively the model can reconstruct the original image when conditioned solely on the prompt derived from that image. Table 2 reports CLIP scores under different feedback

Table 1: Comparison of stylization methods across KID, CLIP, StyleBench, and human evaluation scores. Methods are grouped into fine-tuned, encoder-based, and training-free categories.

| Category | Method | KID ↓ | CLIP Score ↑ | Style Bench ↑ | Human Eval ↑ |
|---|---|---|---|---|---|
| **Fine-tuned** | Textual Inversion | 0.042 | 0.2124 | 0 | 0.5 |
| | DreamBooth | 0.036 | 0.1910 | 0 | 1 |
| **Encoder-based** | CSGO | 0.140 | 0.1977 | 1.5 | 1 |
| **Training-Free** | StyleID | 0.213 | 0.2161 | 1.5 | 1.5 |
| | RF-Inversion | 0.166 | 0.1902 | 1.5 | 2 |
| | RB-Modulation | 0.035 | 0.2069 | 0.5 | 0.5 |
| | DiffArtist | 0.255 | 0.1966 | 1.75 | 0.5 |
| | InstantID | 0.035 | 0.2168 | 1 | 1.5 |
| | **Ours** | **0.025** | **0.2272** | **4** | **3.8** |

Table 2: CLIP scores across feedback conditions. Evaluation uses the same reference for all prompt stages.

| Prompt Type | Vanilla Prompt | Feedback-1 | Feedback-2 | Feedback-3 | Verifier |
|---|---|---|---|---|---|
| CLIP Score | 0.6558 | 0.6980 | 0.7367 | 0.8494 | 0.8575 |

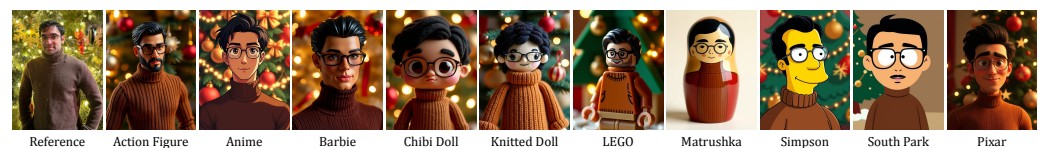

| Reference | Action Figure | Anime | Barbie | Chibi Doll | Knitted Doll | LEGO | Matrushka | Simpson | South Park | Pixar |

Figure 4: **Stylized Generation from text-only Prompts after Identity Distillation.** In this stage, the image is no longer used, only the distilled stylized text prompt is fed to the image generation model. The resulting stylized outputs preserve key identity traits such as hairstyle, clothing, and pose, despite the absence of direct visual reference. This demonstrates the effectiveness of our identity distillation pipeline in guiding style-consistent abstraction purely from text.

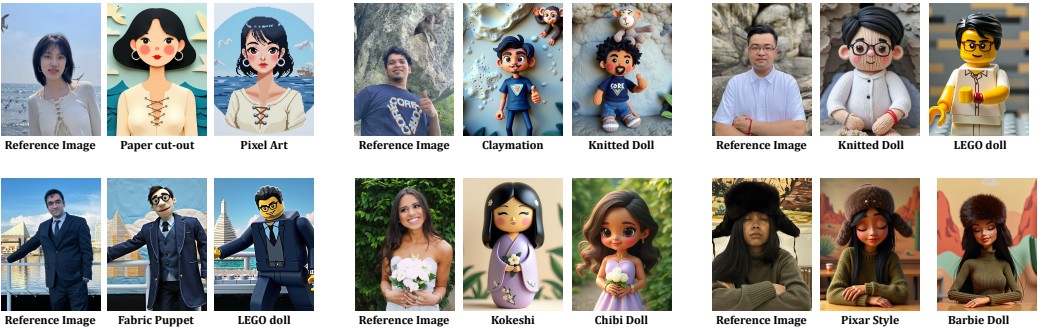

| Reference Image | Paper cut-out | Pixel Art | Reference Image | Claymation | Knitted Doll | Reference Image | Knitted Doll | LEGO doll |
| Reference Image | Fabric Puppet | LEGO doll | Reference Image | Kokeshi | Chibi Doll | Reference Image | Pixar Style | Barbie Doll |

Figure 5: **Additional results across diverse subjects and abstract styles.**

conditions, highlighting the influence of inference-time scaling on identity fidelity. Qualitative examples are shown in Figure 6.

**Impact of Prompt Stylization.** Prompt stylization involves enriching the original prompt with style-consistent descriptors, for e.g., replacing generic phrases with detailed attributes such as "button eyes" or "yarn hair" for knitted dolls, or "yellow skin" for Simpsons-style characters. This guides the model toward more faithful stylistic abstraction. Qualitative differences are shown in Figure 4.

**Impact of Cross-Domain Latent Reversal.** To evaluate the effectiveness of our cross-domain latent reversal framework, we present qualitative comparisons in Figure 7. The source latent reversal baseline starts from the original image and uses a densely styled prompt to directly generate a stylized

| Source Image | ID-v1 | ID-v2 | ID-v3 | ID-v4 | Verifier |
|---|---|---|---|---|---|

Figure 6: **Multi-round inference-time scaling with VLLMs for identity distillation.** At each round, the VLLM extracts identity-relevant features from the original image to reconstruct a refined base representation. This iterative process progressively distills semantic identity (e.g., facial structure, clothing, posture) while filtering out irrelevant details. The final distilled output serves as a robust foundation for stylized abstraction, enabling faithful and expressive generation across diverse styles.

output. However, this approach often fails to capture details such as in the knitted doll example, the facial texture lacks the distinctive knitted pattern due to the absence of a strongly stylized starting point. In contrast, our cross-domain latent reversal begins from an already stylized abstraction, resulting in better preservation of style-specific features. Similarly, in the Barbie doll case, source latent reversal over-emphasizes style at the cost of structural integrity, while our method achieves a more balanced reconstruction of both style and identity.

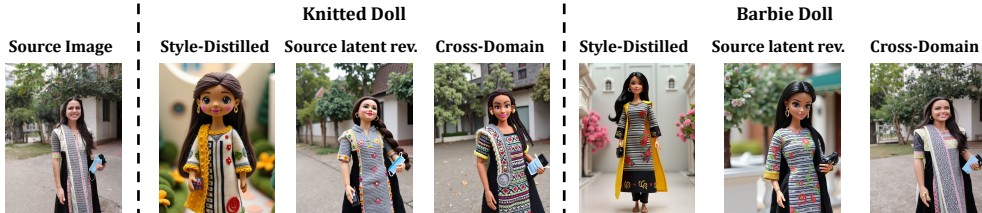

Figure 7: **Effect of Cross-Domain Latent Reversal.** Given a stylized reference and original image, our method uses a VLLM to balance style and structure. Cross-domain reversal starts from a text-initialized latent and iteratively aligns with the reference style while preserving structure. In contrast, source latent reversal starts from the original and applies the style prompt, often disrupting structure. Our approach yields more coherent, identity-preserving abstractions.

# 6 CONCLUSION

We present a training-free framework for stylized abstraction that integrates vision-language inference scaling with cross-domain rectified flow inversion. By dynamically modulating structural restoration using learned style priors, our method enables faithful yet flexible abstraction across diverse stylized domains. Combined with a new evaluation protocol, StyleBench, this work establishes a foundation for abstraction-aware generation of everyday subjects, supporting creative applications without model fine-tuning.

**Limitations** Our method inherits limitations from the underlying VLLM and image generation models, particularly in handling rare styles and edge cases, which may impact output fidelity and generalization. Notably, the model can exhibit racial or cultural biases, such as consistently associating "South Asian" prompts with bindis or traditional jewelry, or "Middle Eastern" with facial hair, regardless of whether these features are present in the reference image. These biases reflect broader challenges in mitigating race-related stereotyping within generative models. To mitigate this, at inference, we apply bias-aware prompt regularization and post-hoc debiasing filters to discourage stereotypical associations when the prompt or reference image does not warrant them.

**Ethical Statement** We use real human images with explicit consent strictly for research purposes. This work supports applications in areas like merchandising, creative content generation, and ideation. Any potential misuse is against our intentions and values.

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
