# Supplementary Materials for Training-Free Stylized Abstraction

## 1 Additional Experimental Details

On average, identity distillation through the VLLM requires four queries, followed by two additional requests to the FLUX model for final image synthesis. The full prompt generation process typically takes about one minute, while FLUX produces the stylized outputs in approximately 1.15 minutes. This runtime is comparable to GPT-5, which averages 2.5 minutes for generating a stylized image.

## 2 More Baseline Comparisons

We have compared our method with other recent methods and shown the results in Figure 1

## 3 Baselines Setup

**RF-Inversion Rout et al. (2025a).** For RF-Inversion experiments, we use the FluxPipeline with pretrained weights from FLUX.1-dev, applying a fixed set of 10 stylized prompts (e.g., Knitted Doll, Lego, Anime, Pixar, Barbie Doll, etc.). Each input image is first inverted into the latent space using 28 inversion steps with a gamma value of 0.5. Stylized outputs are then generated from the inverted latents using the corresponding prompt, with a diffusion range restricted to the first 25% of the process (start timestep=0.0, stop timestep=0.25) across 50 inference steps. We use eta=0.9 for sampling noise and fix the random seed for deterministic results.

**RB-Modulation Rout et al. (2025b).** We evaluate the RB-Modulation framework, which is built on top of StableCascade, for multi-style portrait generation. The system employs a two-stage cascade: Stage-C integrates style guidance using a reference image via the RBM module, while Stage-B decodes the modulated representation into a high-resolution image. For each input image, we condition generation on a fixed set of ten styles, including *Knitted Doll*, *Lego*, *Anime*, *Pixar*, *South Park*, *Barbie Doll*, among others. Style is injected by computing EffNet embeddings from a style reference image, which are passed to Stage-C. Sampling is performed using a cosine schedule with 20 denoising steps for Stage-C (cfg $= 4.0$) and 10 steps for Stage-B (cfg $= 1.1$). The generation prompt follows the format: ``a portrait of a [man/woman] in [style] style''. All images are generated at a resolution of $1024 \times 1024$ and saved under structured directories organized by style. Model weights are loaded from preconfigured files (`stage_c_3b.yaml` and `stage_b_3b.yaml`), and all experiments are run on a single NVIDIA GPU with `bfloat16` precision.

**StyleID Le & Carlsson (2022).** The model performs DDIM inversion of both content and style images to extract intermediate attention features across multiple timesteps and layers. These features are selectively fused by preserving the `query` components from the content and injecting `key/value` components from the style via cross-layer modulation. Images are center-cropped and resized to $512 \times 512$, then encoded into a latent space using a 50-step DDIM inversion process. Feature fusion occurs from step 49 to 0 using layers 6 through 11 of the U-Net architecture. Inference is performed using a modified DDIM sampler with temperature scaling ($T = 1.5$) and query preservation strength ($\gamma = 0.75$), allowing detailed control over style-content disentanglement.

**InstantID Wang et al. (2024).** We employ the InstantID pipeline, a face-aware stylization system built on top of Stable Diffusion XL with ControlNet and IP-Adapter conditioning. For each image, we detect and extract facial features using the InsightFace 'antelopev2' model, which provides both facial landmarks and a 512-dimensional identity embedding. The largest

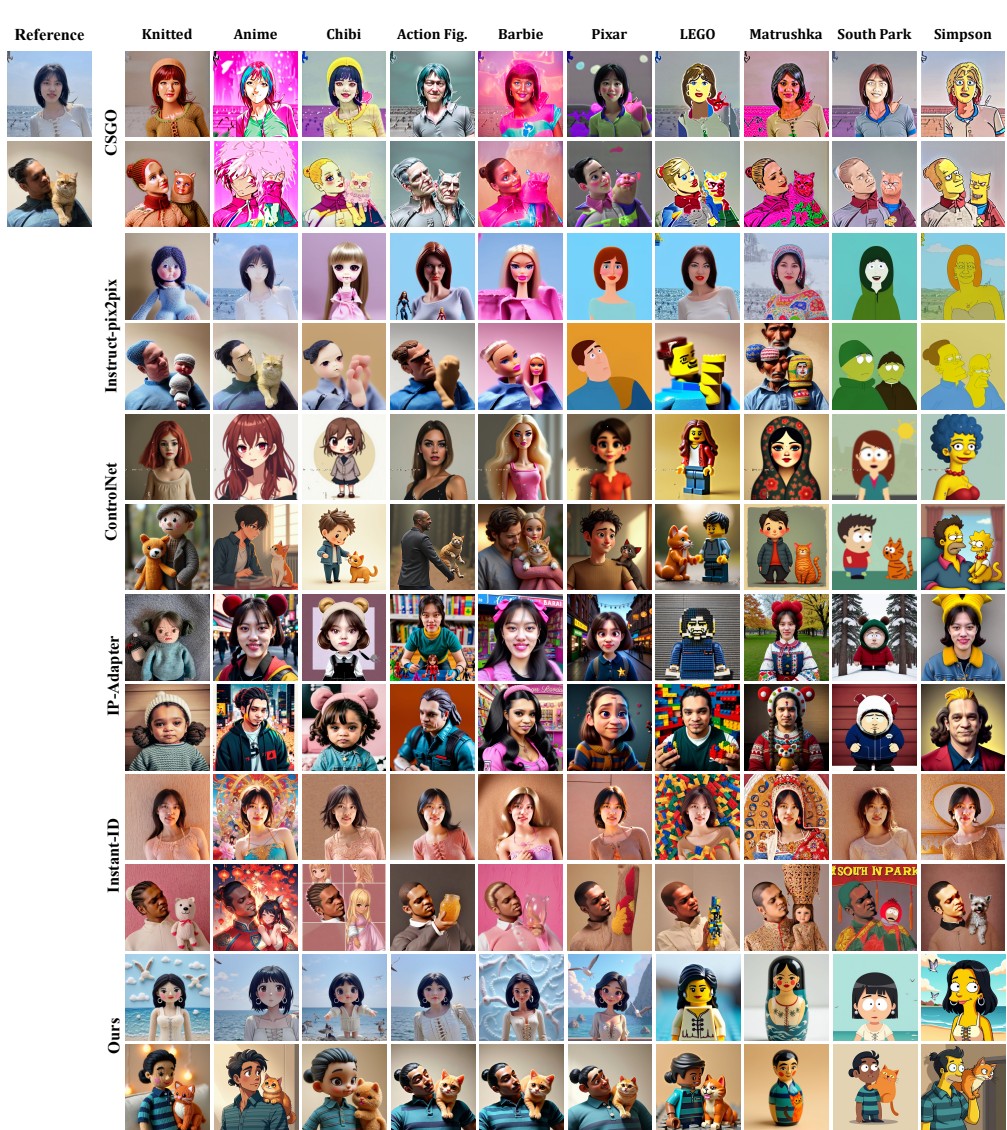

Figure 1: Qualitative comparison with other state-of-the-art methods.

detected face is used for identity preservation. The model is composed of a base SDXL backbone (`stabilityai/stable-diffusion-xl-base-1.0`), augmented with ControlNet and an IP-Adapter module. Facial keypoints are rendered as a control image, while the identity embedding is injected via the IP-Adapter. The prompt used follows the format: ``a portrait of [man/woman] in [style] style'', with gender determined based on pre-mapped image indices. Images are resized to a maximum of $1280 \times 1280$ with alignment to multiples of 64 for efficiency. We apply 30 inference steps with a guidance scale of 5.0, and both the ControlNet and IP-Adapter conditioning scales are set to 0.8. For each image and style (e.g., *Knitted Doll*, *Lego*, *Pixar*, etc.). All experiments are performed on a CUDA-enabled GPU with float32 precision.

**DreamBooth Ruiz et al. (2023).** We fine-tune a Stable Diffusion model using the DreamBooth method for subject-driven generation. The model is trained with a unique identifier token associated with a specific subject. During inference, we use the fine-tuned model to generate stylized outputs conditioned on prompts such as ``a photo of sks person in south park style''.

We utilize the `StableDiffusionPipeline` from the `diffusers` library with half-precision (float16) enabled for memory efficiency. Classifier-free guidance is set to 7.5, and inference is performed with 50 sampling steps. Each prompt is repeated four times to generate a batch of diverse samples. All experiments are conducted on an NVIDIA GPU with CUDA acceleration.

**CSGO Xing et al. (2024).** The model is built on top of Stable Diffusion XL and integrates IP-Adapter modules for content and style injection, in conjunction with ControlNet for structural guidance. We use a curated set of exemplar images representing ten artistic styles (*Knitted Doll*, *Lego*, *Anime*, *Pixar*, *Barbie Doll*, etc.). For each input image, we determine a prompt of the form `''a portrait of [man/woman] in [style]''` based on predefined subject-gender mappings. For multi-subject images, we use a generalized prompt: `''convert into [style] style''`. The pipeline uses pretrained SDXL weights, a customized VAE, and a ControlNet variant trained on tiling patterns. Identity conditioning is injected via content and style tokens (4 and 32 tokens respectively), targeting both base UNet and ControlNet blocks. The content image is resized to align with the model's base resolution and passed through a shared resampler. Stylization is conditioned on a ControlNet scale of 0.6, with guidance and style scales set to 10.0 and 1.0, respectively. Generation is performed over 50 denoising steps, and the seed is fixed to 42 for deterministic outputs.

**DiffArtist Jiang & Chen (2024).** DiffArtist is built on Stable Diffusion and SDXL. We use 50 DDIM inversion and generation steps with a classifier-free guidance scale of 3.5. Attention injection is configured with `share_key=True`, `share_query=True`, `share_value=False`, `share_resnet_layers=[0,1]`, `share_attn=False`, and `share_cross_attn=False`. AdaIN is enabled (`use_adain=True`), and content anchoring is applied during disentangled generation (`use_content_anchor=True`). SDXL runs in `float16` precision, while the VAE operates in `float32`.

**IP-Adapter Ye et al. (2023).** For IP-Adapter Ye et al. (2023) experiments, we adopt the Stable Diffusion v1.5 base model and integrate the lightweight IP-Adapter module with 22M parameters. Each input image prompt is encoded using a frozen OpenCLIP ViT-H/14 image encoder, followed by a projection into 4 learnable tokens. These image features are injected via decoupled cross-attention layers added to all 16 UNet cross-attention blocks. During inference, we set the image prompt weight to $\lambda = 1.0$ and use an empty text prompt for pure image-prompt conditioning. Sampling is performed using DDIM with 50 steps and a guidance scale of 7.5. All image generations are deterministic with fixed seeds, ensuring reproducibility. The model supports multimodal prompts and is compatible with ControlNet and other controllable adapters without additional fine-tuning.

## 4    PROMPT FOR IDENTITY DISTILLATION

**Facial Attributes.**

I will provide you with an image of a human face. Produce a comprehensive, forensic style facial description in a single dense paragraph, detailed enough for accurate human reconstruction or forensic sketching. Describe the individual's apparent biological sex, race or ethnic background, estimated age range, and general physique or body weight appearance (e.g., underweight, average, heavyset, muscular). Detail the skin tone with precision note undertones (cool, warm, neutral), pigmentation, and visible texture such as freckles, moles, scars, or blemishes. For the face shape, specify whether it is oval, round, square, heartshaped, or diamond, and describe the jawline in terms of sharpness, width, and angularity. Include the forehead's height and width, the hairline contour, and whether it is straight, receding, or widow's peak. The cheekbones should be described with respect to height, prominence, and lateral placement. Indicate the chin's form pointed, cleft, square, rounded, recessed, or protruding. For the nose, describe the overall size, bridge contour (e.g., high, flat, concave, convex), nostril flare, tip shape, and symmetry. Then describe the eyes in terms of shape (almond, round, hooded, monolid), tilt (upturned, downturned, straight), size, spacing, iris color, scleral clarity, and the presence or absence of an eyelid crease. Note eyelash length and curl if prominent. Detail the eyebrows' shape (arched, flat, curved), thickness, length, color, grooming style, and position relative to the eye socket. For the mouth, describe lip shape (e.g., heartshaped, bowshaped), fullness, vertical distance to nose and chin, symmetry, the sharpness or softness of the cupid's bow, and overall coloring or pigmentation. Describe the teeth if visible alignment, spacing, color. Include a description of the ears, noting size, angle of protrusion, lobe attachment, and visibility from the front. Describe hair color, texture (straight, wavy, curly, coiled), length, parting, volume, and any dyeing or graying. Finally, include the inter feature ratios, such as the distance from eyes to nose, nose to mouth, mouth to chin, and face width to height, noting vertical and horizontal harmony or asymmetry. Use anatomically precise, observational language as would be found in professional forensic profiling.

**Clothes and Accessories.**

I will provide you with an image of a human subject. Generate a highly detailed, forensic style description of the subject's clothing, accessories, and appearance related adornments in a single continuous paragraph. The description should allow an observer to accurately reconstruct the subject's outfit, materials, and stylistic presentation. Begin with the overall outfit type, including the layering of garments and whether the attire suggests a formal, casual, professional, athletic, cultural, or weather specific context. Describe each clothing item from top to bottom identify the garment type (e.g., jacket, blouse, hoodie, dress), cut and silhouette (e.g., fitted, oversized, cropped, tailored), and the fabric type (cotton, denim, silk, wool, synthetic blend) including texture (smooth, ribbed, coarse, sheer, glossy, matte). Include color, noting primary tones, secondary hues, patterns (e.g., floral, plaid, geometric, logo prints), and fading or distressing. Specify visible closures such as buttons, zippers, drawstrings, or snaps, and describe collars, sleeves, cuffs, hems, stitching, and trim in detail. Mention the fit and drape on the body, whether loose, structured, or body hugging. Describe pants, skirts, or lower garments similarly, including waistband type, length, cut (e.g., tapered, flared, pleated), and material behavior (e.g., stiff, flowy, elastic). Provide precise detail on footwear, covering type (e.g., sneakers, boots, sandals), color, condition (new, worn, scuffed), sole thickness, fasteners, and branding if visible. For accessories, describe all items such as belts, hats, scarves, bags, jewelry, watches, or sunglasses, including their material (leather, metal, plastic), placement, size, color, and design. For jewelry, note shape, stone types, metal color, and position (e.g., left wrist, right earlobe). Mention any logos, emblems, name tags, or inscriptions, their location, style, and legibility. Include hair style related adornments like hair clips, ties, bands, or veils if present. Indicate whether the clothing is clean, wrinkled, tailored, or weathered, and how it interacts with the subject's posture or movement. Use clear, descriptive language with observational precision, suitable for law enforcement or forensic documentation.

**Posture.**

I will provide you with an image of a human subject. Generate an anatomically precise, forensic style description of the subject's pose, body orientation, and posture in a single dense paragraph, suitable for reconstruction in forensic modeling, animation, or figure drawing. Begin by describing the overall stance whether the subject is standing, sitting, leaning, crouching, walking, or in motion and specify the distribution of body weight (e.g., evenly balanced, shifted to one hip, resting on one leg, slouched). Indicate the torso orientation, such as facing forward, three quarter turn, side profile, or twisted at the waist. Describe the spine posture in terms of straightness, curvature, or slouch. Detail shoulder position (level, raised, dropped, angled) and arm placement, specifying whether arms are relaxed, crossed, bent, akimbo, behind the back, or holding an object. Note the hand position, including whether fingers are spread, curled, pointing, or interacting with any surface, accessory, or garment. Include the leg position and angle whether straight, crossed, bent, staggered, or one knee slightly raised and describe foot placement relative to the body and ground, noting if feet are parallel, angled outward/inward, or midstride. Specify the head tilt, pitch, yaw, and roll (e.g., upright, nodding forward, tilted to the side, turned partially), and describe eye gaze direction (e.g., looking straight ahead, offcamera, downward, or toward an object or person). Indicate whether the pose appears relaxed, tense, alert, casual, formal, or dynamic, and if the subject is interacting with the environment (e.g., leaning against a wall, sitting on a chair, walking up stairs). Include relational geometry such as angles between limbs, distance between hands and torso, and chin to shoulder alignment to allow for 3D pose reconstruction. Use clear anatomical and kinesiological language appropriate for forensic modeling or biomechanical analysis.

**Background.**

I will provide you with an image containing a human subject within a visible environment. Generate a comprehensive, continuous paragraph that describes the background and setting in exhaustive detail, suitable for recreating the scene in forensic reconstruction, virtual staging, or cinematic layout. Begin with the type of environment whether it is indoor or outdoor, public or private, residential, commercial, natural, or constructed. Describe the primary spatial context such as a street, room, park, studio, hallway, or landscape, and specify depth cues like perspective, vanishing points, and visible horizon lines. Detail the lighting conditions, indicating whether the light is natural or artificial, the light source direction (e.g., overhead, frontal, backlit), shadow presence and length, and overall ambience (e.g., bright, dim, moody, clinical). Provide the color palette of the background dominant hues, gradients, saturation, and color temperature (warm, cool, neutral). Describe surface textures and materials walls, floors, or ground surfaces (e.g., tiled, carpeted, wooden, concrete, grassy), their cleanliness, reflectiveness, or damage. Include any visible structures or objects, such as furniture, windows, fences, signage, artwork, cables, curtains, vehicles, trees, or architectural details, describing their placement, size, condition, and interaction with light or shadows. Mention background activity if any other people, movement, animals, or vehicles and whether the environment is static or dynamic. Indicate the depth of field how blurred or sharp background elements appear relative to the subject. Include information about weather or atmospheric effects (e.g., fog, rain, sunlight, haze, reflections), time of day, and seasonal indicators (e.g., dry leaves, snow, blossoming plants). Describe any visible text, logos, posters, or signage in the background, noting legibility, language, and style. Conclude with the spatial relationship between the subject and background how far the subject appears from walls, objects, or vanishing points and whether there are any visual elements framing or isolating the subject. Use precise, observational language suitable for forensic, cinematic, or architectural analysis.

**Combine.**

I will provide a detailed description of an image. Your task is to compress and distill the description into two versions:

**T5 Embedding Prompt (512 tokens max):** Use all 512 tokens. Preserve the most salient, semantically rich, and distinctive features of the image that would help T5 encode identity, context, style, and layout. Discard irrelevant or negative information (e.g., "no visible tattoos") if necessary for brevity.

**CLIP Embedding Prompt (77 tokens max):** Create a compressed caption and use all 77 tokens, optimized for CLIP's contrastive text-image embedding. Prioritize visual discriminability and identity-relevant details, especially those that can anchor visual features (e.g., colors, shapes, facial attributes, clothing style, environment cues).

In both cases, your goal is to retain the most distinguishing and generative elements from the input description, suitable for retrieval or conditional generation tasks.

## 5 STYLEBENCH EVALUATION PROMPT

**Prompt.**

**Task Definition** You will be provided with three inputs: a reference image, a stylized generated image, and a style prompt. Your task is to evaluate how well the generated image captures the intended style abstraction while preserving the recognizable identity of the subject in the reference image.

**Evaluation Focus** This is a task of stylized abstraction, not realism or direct replication. The goal is not to retain exact facial proportions or textures from the reference but to abstract the subject into a distinct artistic or toy-like style. The reference image provides the identity cues, such as hairstyle, accessories, clothing, skin tone, or posture, which should be recognizable in the generated image despite heavy stylistic transformation. The style prompt dictates the visual language that the generation should adhere to, such as the yarn texture of a knitted doll, the plastic blockiness of a LEGO figure, or the yellow skin and cartoon geometry of a Simpson character. A good abstraction interprets the identity through the lens of the target style. For instance, in a LEGO doll style, the facial features may become minimal and geometric, but an iconic element like Einstein's wild hair or Michael Jackson's fedora must still be present. In a South Park style, the flat shading and round cut-out forms are expected, while the individual identity is retained through clothing or hair color. Likewise, in Ghibli or Van Gogh styles, the abstraction involves painterly texture and expressive strokes rather than exact geometry. The fidelity of abstraction lies in this balance between stylization and recognizability.

**Scoring Criteria** You must assess the generated image based on three integrated dimensions:

- How well the image adheres to the specified style in the prompt.

- Whether key identifying features from the reference image are preserved.

- Whether the abstraction effectively fuses identity and style into a coherent result.

Note that expressions, pose, and compositional cues should come from the reference image. The quality of the abstraction depends on both a strong stylistic transformation and the faithful reinterpretation of subject identity.

**Prompt.**

**Scoring Range** Assign a score from 0 to 4: Very Poor (0): The generated image fails to apply the intended style and does not retain recognizable identity. Poor (1): The style is weak or incorrect; most identity traits are lost. Fair (2): The style is moderately applied, with partial identity preservation. Good (3): The image reflects the correct style and retains most identity features. Excellent (4): The image is a faithful and expressive stylization that captures both the identity and style seamlessly.
**Input Format** You will receive:
A reference image
A stylized generated image
A style prompt
**Output Format**
**Score: [Your Score]**
Only return the score. Do not include any justification or explanation.