# OpenReview forum: "Training-Free Stylized Abstraction"
_ICLR.cc/2026/Conference — Submitted to ICLR 2026_

### Official Review · Reviewer_DUXZ · 2025-10-28

**Soundness:** 3
**Presentation:** 3
**Contribution:** 2
**Rating:** 6
**Confidence:** 4

**Summary:**

This paper introduces a training-free framework for stylized abstraction that leverages inference-time scaling in VLLMs and a novel cross-domain rectified flow inversion to generate identity-faithful yet highly stylized representations from a single image.

**Strengths:**

The core Innovations are as follows:

1. Training-Free Identity-Preserving Abstraction: A novel framework that utilizes inference-time feature scaling in Vision-Language Models (VLLMs) to extract and preserve critical identity cues without any fine-tuning.

2. Cross-Domain Rectified Flow Inversion: An innovative inversion strategy that reconstructs semantic structure by leveraging style-dependent priors, effectively bridging the domain gap between realism and abstraction.

3. Dynamic Style-Aware Reconstruction: A flexible temporal scheduling mechanism that adaptively controls structural restoration based on style intensity, enabling high-fidelity results across varying abstraction levels.

4. Specialized Evaluation Metric: The introduction of StyleBench, a GPT-based human-aligned evaluation metric specifically designed to assess abstract stylization where traditional pixel-level metrics fail.

**Weaknesses:**

1. The human evaluation setup for StyleBench is a weakness. The assessment of only 25 generated images is too small a sample size to robustly validate the metric's alignment with human judgment. Furthermore, the methodology lacks crucial details on how annotations from the 15 evaluators were aggregated, particularly in cases of disagreement, which undermines the reliability of the reported results.

2. The framework's capability to preserve fine-grained identity attributes appears limited. As observed in Figure 3, the results for styles like 'knitted', 'anime', and 'chibi' show substantial identity divergence from the reference. This raises concerns about the effectiveness of the VLLM in parsing and retaining subtle but critical identity features (e.g., facial moles).

3. There is a potential factual inaccuracy in the presentation: the image for the 'Barbie doll' style in Figure 7 appears to be incorrect, as it does not visually match the stated style, which could mislead readers.

**Questions:**

1. Regarding the human evaluation, could you specify the exact protocol for processing the annotations from the 15 evaluators? For instance, was a measure of inter-annotator agreement (e.g., Cohen's Kappa) calculated, and how were final scores derived from potentially divergent ratings?

2. What is the granularity of the attributes that the VLLM-based parser can reliably extract? Can it capture very fine details, such as specific facial marks, and if not, how does this limitation impact identity preservation for out-of-distribution subjects?

3. Regarding the cross-domain rectified flow inversion, are there any constraints on the selection of the stylized input? How does the method perform when there is a very large domain gap between the real reference image and the chosen stylized input?

---

### Official Review · Reviewer_fZo6 · 2025-10-29

**Soundness:** 2
**Presentation:** 3
**Contribution:** 2
**Rating:** 4
**Confidence:** 4

**Summary:**

This paper defines a new task in the domain of image transfer called stylized abstraction. The authors first proposes a training free pipeline that distills identity with an inference time VLLM loop that produces T5 and CLIP prompts which is fed iteratively into a t2i generation image generation model, and a cross domain rectified flow inversion with temporal scheduling to inject structural fidelity. They also introduce stylebench, a GPT assisted evaluation method for abstraction. The results show qualitative results that adhere to the coined term "stylized abstraction".

**Strengths:**

- The authors has a clear problem framing that distinguishes abstraction from classic style transfer, where they preserve identity while allowing large style driven distortions.
- Their proposed stylebench benchmark explicitly targets abstraction (style adherence + identity + fusion), and also includes the classic KID/CLIPscore to show that these classic metrics are positively correlated with their proposed stylebench benchmark. Including a human preference study in qualitative-heavy work is also appreciated.

**Weaknesses:**

- The main contribution of this work is less technical nor theoretical, but heavily leans into the domain of prompt engineering. The qualitative results do look impressive and I do agree with the authors' formulation of the new task of "stylized abstraction", in order for a work of this nature to be given a high rating the contributions has to go beyond ad-hoc tips and needs to show generality; automation, theory, or rigorous evaluation.
- The main contribution of the proposed workflow seems to be model independent firsthand, but authors do not show any other ablation studies with different models. It is noted that most open source VLM models do use either T5 or CLIP based encoders, but the proposed method hardcodes the number of tokens into their main text (e.g. 77 for CLIP), which limits the generality of the method.
- There is limited to no analyses explaining why certain parts of their methods work, either on an engineering or theoretical level. There is no ablation study that isolate each component's contribution that go beyond select figures/tables.

**Questions:**

- any additional ablations showing the strength of the proposed method (e.g. the select choice of attribute extraction and prompt compression) isolated from the given backbone models (InternVL/Flux) would be appreciated.

---

### Official Review · Reviewer_jPno · 2025-10-31

**Soundness:** 2
**Presentation:** 1
**Contribution:** 2
**Rating:** 2
**Confidence:** 4

**Summary:**

This paper introduces a training-free framework for stylized abstraction. This task aims to generate stylized yet identity-preserving representations of individuals in highly abstract visual styles. The method first distills identity-relevant semantic attributes from the input image using VLLM. Then it employs a cross-domain latent inversion process based on rectified flows to reconstruct stylized representations while preserving core identity cues. The authors further propose StyleBench, a GPT-based evaluation protocol that measures abstraction quality more effectively beyond pixel-space similarity.

**Strengths:**

- The paper introduces stylized abstraction as a meaningful application task.
- Introducing StyleBench addresses an evaluation gap in stylization tasks, where pixel-based metrics are not well-suited.
- The qualitative results are compelling, demonstrating visually coherent abstractions across diverse styles and subjects.

**Weaknesses:**

- The identity distillation step is largely prompt engineering, while the stylization and reconstruction pipeline is built on RF-Inversion. The contribution feels more like combining existing components than introducing new algorithmic insights.

- The method explanation is difficult to follow. Figure clarity and pipeline grounding are lacking. For example, in Figure 2, the transition between generated outputs and stylized inputs is not clearly explained.

- The computational cost appears significantly higher compared to alternative training-free personalization/editing approaches, but no runtime or resource comparison is provided.

**Questions:**

- The overall method explanation is difficult to follow. In particular, Figure 2 does not adequately convey the whole pipeline. For example, in Fig. 2(a), the output of the generation stage becomes the stylized input in Fig. 2(b), but this transition is not explicitly described. Providing a more step-by-step procedural diagram or pseudocode would significantly improve clarity.

- The evaluation would be stronger with comparisons to recent open-source T2I editing models, such as Flux-Kontxt or Qwen-Image-Edit. Additionally, even if the performance differs, including comparisons to GPT or Gemini outputs would help contextualize the practicality of the method in real-world creative workflows.

- The proposed pipeline relies on iterative prompt refinement and multiple VLLM interactions, which likely increases computational overhead compared to prior training-free approaches. Therefore, it would be essential to include a runtime and resource comparison, especially against training-free baselines, to assess practical feasibility.

- A failure case analysis would also be valuable for understanding the limitations of the proposed method. For instance, if the T2I model fails to preserve identity even when the textual description is accurate, how does the pipeline react? Explicit discussion of such scenarios would clarify the strengths and limitations of the approach.

---

### Official Review · Reviewer_Kva8 · 2025-11-03

**Soundness:** 3
**Presentation:** 4
**Contribution:** 2
**Rating:** 4
**Confidence:** 3

**Summary:**

In this paper, the authors define the task of stylized abstraction, that is generating images that exaggerate or simplify appearance while keeping a target subject recognizably the same. This is distinct from classic style transfer, which aims to significantly retain structure. The introduced task is very interesting and the authors also propose a dedicated benchmark/evaluation protocol (StyleBench).

To solve this task, the authors propose a training-free pipeline that uses multiple V(L)LMs, foundation models like CLIP, and rectified-flow models such as FLUX. First, the authors distill the subject’s identity into natural language by prompting a VLLM multiple times. Next, they regenerate images based on these descriptions and compare them with the original image until a CLIP threshold is reached. Finally, they apply a style-aware prompt transformation to produce stylized prompts.

In the second stage, the authors invert the stylized image using cross-domain rectified-flow inversion with a style-aware temporal schedule, selectively restoring structure while preserving the intended style.

The results, both quantitative and qualitative, show that the method performs significantly better than traditional style transfer pipelines on the task of stylized abstraction.

Overall, the method is elegant and the task is interesting; however, I’m unsure whether the heavy usage of pretrained models delivers sufficient methodological novelty for ICLR.

**Strengths:**

* The task introduced is interesting. It clearly distinguishes stylized abstraction from classic style transfer and motivates why it matters.

* The presentation of the paper is also nice and clear. The paper is easy to follow, and the visual results strongly support the authors’ claims.

* The method, although it leverages many off-the-shelf models, is intuitive and elegant. The authors also show ablation studies on the effect of cross domain latent reversal, the multi-turn identity distillation and prompt stylization.

* The authors also contribute a benchmark for stylized abstraction, albeit limited.

**Weaknesses:**

1. The evaluation set and human study are pretty narrow to support strong claims. Only 15 people where used on 25 generated images.

2. While the pipeline is elegant, the contribution to ICLR is not very clear, since the authors mainly make usage of large pretrained models.

3. The heavy usage of LVLMs likely has a high inference cost. It would be good if the authors actually mention runtimes.

**Questions:**

* What is the inference cost of the method? How long does it take to obtain one final result ?
* What are the limitations of your method ?
* Will the stylebench benchmark be made available for other researchers ?

Overall, my main concern with the paper is that it is mostly a composition of pretrained models and the novelty for ICLR is not apparent, considering also the limited size of the experimental dataset. I would like the authors to comment on this.

---

### Meta-Review · Area_Chair_PLd8 · 2026-01-03

**Summary:**

The reviewers raised several consistent concerns, and the final decision was primarily based on the following points.

First, the evaluation is considered weak. On the one hand, the human study is too limited to support strong claims, as it relies on only 15 evaluators assessing 25 generated images (Kva8, DUXZ). On the other hand, the paper lacks sufficient details on how annotations were aggregated or how disagreements among evaluators were handled, which undermines the reliability and robustness of the reported alignment between the proposed metric and human judgment (DUXZ).

Second, the contribution is viewed as insufficient relative to ICLR standards (Kva8, jPno, fZo6). While the paper introduces the notion of stylized abstraction and presents visually appealing qualitative results, reviewers felt that the technical novelty is limited. The proposed framework largely combines existing components (e.g., RF-Inversion and pretrained LVLMs) and relies heavily on prompt engineering, with limited new algorithmic or theoretical insights.

Third, the computational cost of the framework raises concerns (Kva8, jPno). The heavy reliance on large vision-language models likely results in substantial inference overhead, yet the paper does not provide runtime, efficiency, or resource comparisons with alternative approaches. Additionally, the method description is challenging to follow, with unclear explanations of the pipeline and figure transitions, which further impairs the clarity and accessibility of the work.

**Reviewer Concerns:**

All concerns are still outstanding as there is no rebuttal provided.

**Reviewer Scores:**

N/A, as there is no rebuttal provided.

---

### Decision · Program_Chairs · 2026-01-26

Reject